# The Effect of Capital Structure on Firm Value: A Study of Companies Listed on the Vietnamese Stock Market

**Thi Ngoc Bui [1], Xuan Hung Nguyen [2,*] and Kieu Trang Pham [2]**

1    Faculty of Economic and Management, Thuyloi University, No. 175, Tay Son, Dong Da,
     Hanoi 100000, Vietnam; ngocbt_kt@tlu.edu.vn
2    National Economics University, 207 Giải phong, Hai Ba Trung, Hanoi 100000, Vietnam;
     11226479@st.neu.edu.vn
*    Correspondence: hungnx@neu.edu.vn

**Abstract:** This research investigates the relationship between capital structure and firm value for companies listed on the Vietnamese stock market. The study utilizes data from audited financial statements of 769 companies spanning from 2012 to 2022, amounting to 8459 observations. Employing various estimation methods, such as ordinary least squares (OLS), fixed effects model (FEM), random effects model (REM), and generalized least squares (GLS), the impact of capital structure on key financial indicators, namely, return on assets (ROA), return on equity (ROE), and Tobin's Q, is assessed. The findings indicate that the debt ratio exhibits a positive influence on ROA, ROE, and Tobin's Q, with Tobin's Q displaying the most pronounced impact (0.450) and ROA showing the weakest impact (0.011). However, the long-term debt ratio does not significantly affect firm value. Interestingly, both short-term and long-term debt ratios have negative effects on ROA, ROE, and Tobin's Q, with the most substantial impact on Tobin's Q reduction (0.562). Based on these research outcomes, the authors offer valuable recommendations to companies, investors, business leaders, and policymakers to make informed decisions in selecting an optimal and sensible capital structure.

**Keywords:** ROA; ROE; Tobin's Q; capital structure; firm value; effect

## 1. Introduction

The intricacies surrounding a company's capital structure make it a highly debated and intricate subject globally, particularly when examining its potential impact on the enterprise's profitability and overall value. As the stock market in Vietnam operates within a distinct economic environment, shaped by unique socio-political institutions and a specific financial-banking system, conducting a study on the influence of capital structure on firm value will provide valuable empirical insights to enrich the capital structure theory.

Capital structure, also referred to as finance leverage or financial structure, encompasses various terms and is commonly known as capital structure or financial leverage. It signifies the proportion of debt and equity utilized to fund a business's asset formation. The level of debt employed has a significant impact on managerial behavior and financial decision making. Financial ratios, including the debt/equity ratio (long-term debt/equity or long-term debt/total capital employed), provide a means to gauge the capital structure within a company. Moreover, the authors contend that the capital structure ratio can include both short-term and long-term debt, particularly when a business has an extended overdraft. According to (Watson and Head 2007; Khan and Jain 1997), capital structure represents the blend of debt and equity used by a firm to finance its long-term business operations. Capital is viewed as a long-term source of financing in an enterprise and is determined by subtracting short-term liabilities from total assets. The allocation of the total business value between creditors/debtholders/bondholders and owners/shareholders/equityholders is showcased through the capital structure.

The evaluation of firm value encompasses key indicators, such as return on equity (ROE), return on total assets (ROA), and Tobin's Q. Efficient businesses effectively utilize capital, leveraging tax shields to their advantage. In contrast, inefficient businesses with low competitiveness and mounting debts that increase bankruptcy risk are compelled to reduce their debt ratio. The inefficient use of capital structure, particularly excessive debt, results in a contrasting effect known as financial leverage. Consequently, firms adopt diverse capital structure strategies in response to their unique situations. Among various factors influencing firm value, capital structure holds a crucial influence. Empirical studies examining the impact of capital structure on firm value have yielded mixed conclusions. Some studies on capital structure demonstrate that firms with higher debt ratios often exhibit improved business performance, aligning with the findings of (Berger and Bonaccorsi Di Patti 2006; Weill 2008; Chowdhury and Chowdhury 2010; Nguyen et al. 2023; Khan et al. 2021; Ayuba et al. 2019; Natsir and Yusbardini 2020; Ramli et al. 2019; Hirdinis 2019; Nenu et al. 2018; Aggarwal and Padhan 2017). This result aligns with Modigliani and Miller's (1963) tax shield theory, suggesting that debt creates a tax advantage that maximizes firm value. However, other studies, particularly in developing countries, such as Jordan, Ghana, South Africa, and India, have found a negative relationship between capital structure and business performance, as observed in the works of (Zeitun and Haq 2015; Dawar 2014; Nguyen et al. 2023; Dang et al. 2019). These studies indicate that higher debt levels increase bankruptcy risk, thus reducing firm value in emerging economies. In the context of Vietnam, empirical research highlights the detrimental impact of a high debt ratio on corporate profitability. Additionally, the studies of (Đ. B. Thanh 2016) affirm a close association between capital structure and enterprise value. Nonetheless, the variability in empirical findings underscores that this relationship depends on the economic context, the methods of recording financial indicators, and the adopted research methodologies.

This study seeks to analyze the impact of capital structure on firm value within companies listed on the Vietnamese stock market during the period 2012–2021, offering insights into identifying an optimal capital structure that positively influences firm value. The paper is organized as follows: (1) Introduction; (2) Theoretical framework; (3) Research overview; (4) Hypotheses, research model, and data; (5) Research results and discussion; and (6) Conclusions and recommendations.

## 2. Literature Review and Theoretical Framework

### 2.1. Literature Review

#### 2.1.1. Research on Capital Structure

In their studies, (Khan and Jain 1997; Demirguc-Kunt et al. 2019) define capital structure as the composition of debt and equity utilized by businesses to finance long-term operations. Capital is perceived as a long-term financing source in an enterprise, determined by deducting short-term liabilities from total assets. According to (Watson and Head 2007), the capital structure signifies the proportion of liabilities relative to the equity portion adopted by the business. Various financial indicators, such as debt/equity ratio, long-term debt/equity ratio, long-term debt ratio to total capital employed, and current and long-term debt ratio/equity, serve as measures to assess the capital structure within the enterprise. Essentially, the capital structure reflects the division of the total enterprise value between creditors and owners (shareholders). Additionally, capital structure evaluation can be based on the book value or market value of assets, liabilities, and equity. Liabilities play a vital role in determining the enterprise's total debt, distinguishing between long-term debt, short-term debt, and loan debt. (Ross et al. 2013) argue that a company's capital structure is the amalgamation of debt and equity which is maintained by the organization. When seeking external capital to finance projects, businesses must decide on the type of securities to be issued and establish the appropriate capital structure. For capital structure analysis, the authors employ the total debt (comprising short-term and long-term debt)/equity indicator.

Concepts of optimal capital structure: The notion of optimal capital structure, as discussed by (Davidson 2018; Watson and Head 2007), suggests that there exists a specific financing ratio at which a business can derive maximum benefits. To achieve this optimal capital structure, enterprises must make appropriate adjustments. It is attained when companies strategically coordinate their capital sources to yield the highest advantages. At the point of optimal capital structure, the enterprise experiences the lowest cost of capital, maximizing the value of the owners' assets. Determining the optimal amount of outstanding debt is crucial to establish this ideal capital structure. Various market factors, such as taxes, financial constraints, agency costs, and asymmetric information, influence the attainment of the optimal capital structure. Indicators reflecting the capital structure include:

First, the debt-to-total assets ratio, an essential indicator reflecting the capital structure, showcases the proportion of assets financed through borrowed capital. It illuminates the extent of a company's reliance on borrowed capital sources. This ratio can be calculated using the total debt, long-term debt, or short-term debt, each corresponding to the ratio of total debt, long-term debt, and short-term debt-to-total assets. A low debt-to-total assets ratio indicates a high level of financial independence from creditors, suggesting less dependence on debt. Conversely, a higher ratio signals a greater likelihood of debt payment difficulties or the risk of bankruptcy.

Second, the debt-to-equity ratio provides insights into the balance between the two primary capital sources utilized by businesses to fund operations, namely, debt and equity. This ratio serves as a reflection of the company's capital structure. Similar to the debt-to-total assets ratio, the liabilities in the numerator can be represented by total debt, long-term debt, or short-term debt, depending on the analytical purpose.

### 2.1.2. Research on Firm Value

(Venkatraman and Ramanujam 1986) illustrate that firm value can be gauged through various financial metrics, including revenue growth, profitability, earnings per share, market value/book value ratio, and Tobin's Q. Additionally, non-financial indicators, such as brand recognition, market share value, product quality, marketing efficiency, and production value, are used to measure firm value. Scholars including (Ali et al. 2022; Mihail and Micu 2021; Phillips and Sipahioglu 2004; V. D. Thanh 2016) have focused on the target of return on assets (ROA) as an important indicator of firm value. Conversely, (Dahmash et al. 2023; Nguyen et al. 2023; Abor 2005; Yang et al. 2010; Ebaid 2009) have emphasized the significance of return on equity (ROE) in assessing firm value. Furthermore, (Jiraporn and Liu 2008; Vo 2017) have examined the impact of a stock's market price using Tobin's Q indicator. Therefore, in this study, we will employ three key indicators, namely, ROA, ROE, and Tobin's Q, to assess business value.

Return on assets (ROA) is derived by dividing the net return after taxes by the average assets. This essential indicator gauges how much profit is generated from each dollar of assets after accounting for taxes. ROA serves as a comprehensive measure, reflecting the business's profitability, operational status, and efficiency. Notably, ROA assesses a company's ability to generate profits before incorporating the impact of financial leverage. In contrast to ROE, ROA encompasses all assets, including those financed by both debts and investor contributions. As a result, ROA offers insights into the company's effectiveness in managing and utilizing assets to generate profits. A higher ROA signifies the business's superior capability in generating profits from its assets, making it an indicator of profitability and productivity efficiency, as highlighted by (Phillips and Sipahioglu 2004; Wong et al. 2021).

Return on equity (ROE) is computed as the net return after taxes divided by the average equity. This crucial metric reveals how much profit is earned from each dollar of equity after considering taxes. ROE serves as a reflection of the business's operational status and the efficiency with which the invested capital is managed and utilized. In essence, it signifies the effectiveness of corporate governance in generating profits for shareholders. A

higher ROE indicates enhanced profitability and a strong business value, serving as a basis for assessing managerial competence. If the ROE falls below the prevailing interest rate, shareholders may lack the incentive to invest further in the business. Notably, ROE stands for one of the primary financial indicators widely employed in evaluating corporate value, as underscored by (Yang et al. 2010; Ebaid 2009).

Tobin's Q is determined by the formula (capitalization of common stock + market value of debt + market value of preferred stock) divided by the book value of total assets. However, ascertaining the market value of debt proves challenging due to the varied forms, maturities, and interest rates through which businesses secure loans. Addressing this, (Brainard and Tobin 1968) argued that the capitalization of common stock primarily influences corporate value, allowing for the market value of debt and preferred stock to be replaced with the book value of debt and preferred stock or the value of total liabilities. Consequently, the Tobin's Q measurement is calculated as the market value of the enterprise divided by the book value of the enterprise. Tobin's Q relies on both current and expected future profits from market capitalization. A higher Tobin's Q value indicates greater potential and development prospects for the enterprise. Tobin's Q is >1 when the market value exceeds the book value, signifying that the return on equity surpasses the required rate of return. Conversely, when the return on equity falls below market demands, Tobin's Q is <1, as highlighted by (Jiraporn and Liu 2008).

### 2.1.3. Research on the Impact of Capital Structure on Corporate Value

Presently, numerous research endeavors worldwide have investigated the influence of capital structure on enterprise value. However, the variability in sample sizes, diverse business sectors, and geographical locations of the studied companies has led to disparate findings. In this study, we categorize the research into three distinct cases:

First, capital structure has no impact on corporate value: (Jiraporn and Liu 2008) studied 1900 listed companies between 1990 and 2004 and concluded that capital structure does not affect corporate value. While (Ebaid 2009) studied companies in Egypt, (Phillips and Sipahioglu 2004) studied 43 companies listed on the UK stock market and came to the same conclusion.

Second, capital structure has a positive impact on corporate value: (Nguyen et al. 2023) conducted a study on Vietnamese listed companies during the period from 2012 to 2018 regarding the correlation between capital structure and firm profitability. Surprisingly, their findings indicated that short-term loans also had a positive impact on the firm's profitability in the context of Vietnam. This contradicted some previous research, which had suggested that higher leverage leads to better firm profitability. In contrast, studies by (Jin and Xu 2022; Khan et al. 2021; Ayuba et al. 2019) demonstrated that an increase in the percentage of debt in the capital structure positively influences firm value. (Natsir and Yusbardini 2020) analyzed financial statements from 17 public companies and found significant effects of firm size and capital structure on profitability and firm value. Furthermore, they observed that profitability acted as a mediator in the relationship between firm size, capital structure, and firm value. In a different context, (Ramli et al. 2019) investigated the impact of capital structure determinants on firm financial performance in Malaysia and Indonesia during the period from 1990 to 2010. Their results revealed that firm leverage played a mediating role in Malaysia but not in the Indonesian sample. The attributes indirectly influenced by firm leverage on firm financial performance included asset structure, growth opportunities, liquidity, non-debt tax shield, and interest rates. In another study by (Hirdinis 2019), capital structure was found to have a significant positive effect on firm value, while firm size had a significant negative effect on firm value. Surprisingly, profitability showed no significant effect on firm value, but company size had a significant positive effect on profitability. (Nenu et al. 2018) investigated the Romanian market (companies listed on the Bucharest Stock Exchange) from 2000 to 2016 and found that leverage positively correlated with company size and share price volatility. However, the impact of debt structure on corporate performance varied depending on whether accounting measures or market share

price evolution were considered. Finally, Aggarwal and Padhan 2017 studied the listed Indian hospitality firms over the period from 2001 to 2015. Their findings revealed a significant relationship between firm value and firm quality, leverage, liquidity, size, and economic growth.

Chowdhury and Chowdhury (2010) conducted a study on the impact of capital structure on corporate value in Bangladesh. They collected data from 77 non-financial enterprises listed between 1994 and 2003 and developed a research model with several control variables, including income per share, dividend payout ratio, state shareholding ratio, fixed asset turnover, short-term payment coefficient, operating leverage, revenue growth rate, and enterprise size. The research revealed a positive effect of capital structure on corporate value. Similarly, (Sivathaasan et al. 2013) investigated the factors influencing the profitability of all manufacturing companies listed on the Colombian Stock Exchange in Sri Lanka between 2008 and 2012. The influencing factors included asset structure, capital structure, company size, and growth rate, while the dependent variables representing profitability were ROE and ROA. The study found that only capital structure had a positive impact on companies' profitability, with asset structure variables, company size, and growth rates showing no statistically significant relationship with ROE and ROA. In another study, (Khidmat and Rehman 2014) examined nine chemical companies listed on the Pakistan Stock Exchange between 2001 and 2009. They used ROA as the dependent variable representing business profitability and analyzed factors, such as quick solvency ratio, short-term solvency ratio, debt-to-equity ratio, and debt-to-total asset ratio. The results of the regression model analysis indicated that solvency ratios had a similar impact, while the remaining factors had a negative impact on the profitability of the businesses. These findings align with conclusions from similar studies conducted on listed enterprises in Vietnam, such as the research of (Tran 2016; Đ. B. Thanh 2016; and Vo 2017). Moreover, foreign studies, such as those conducted by (Titman and Wessels 1988; Friend and Lang 1988), support the same viewpoint.

Third, capital structure has a negative impact on corporate value: Several empirical studies provide support for the perspective that capital structure has a negative effect on corporate value. (Vu Thi and Phung 2021; Masulis 1983; Singh and Faircloth 2005) argue that high borrowing rates lead to reduced future investments, thereby negatively impacting the business's value and growth potential. Similar conclusions were drawn in the research conducted by (Seetanah et al. 2014; Dawar 2014; Zeitun and Haq 2015). In a study by (Bolek and Wilinski 2012) on the profitability of construction companies listed on the Warsaw Stock Exchange between 2000 and 2010, it was found that company size and GDP growth had a positive impact on business profitability (ROA), while asset structure, capital structure, average collection period, and quick solvency ratio had the opposite effect. Similarly, (Minnema and Andersson 2018; Owolabi and Obida 2012; Lazaridis and Tryfonidis 2006; Liargovas and Skandalis 2008) discovered a positive relationship between uptime and the scale of operations with business profitability, while the debt ratio had a negative impact. Moreover, (Nguyen et al. 2023; Dang et al. 2019) confirmed a negative relationship between long-term debt and profit maximization. These findings collectively indicate that certain capital structure factors can indeed have adverse effects on corporate value and profitability.

Several other studies have presented both concurrence and contradiction regarding the effects of capital structure on business performance, e.g., (Abor 2005). Utilizing the ordinary OLS method, the study revealed a positive impact of short-term debt and an adverse influence of long-term debt on enterprise value. These mixed results highlight the complexity of the relationship between capital structure and business performance, prompting the need for further investigation and understanding of the underlying factors that contribute to these outcomes.

### 2.2. Some Related Background Theories

#### 2.2.1. Agency Cost Theory

The concept of agency costs in the modern firm model was introduced by (Berle and Means 1932), and later developed into the agency cost theory by (Jensen and Meckling 1976). According to this theory, agency costs arise from conflicts of interest between shareholders, managers, and creditors. These conflicts may lead managers to prioritize personal gains over maximizing the business's value. As a result, managers might choose investment projects with lower risk, lower return, and a reduced debt ratio to minimize the likelihood of bankruptcy. To address these conflicts and reduce agency costs, (Harris and Raviv 1991) proposed the use of debt as a mechanism to monitor and incentivize the performance of the board of directors. This is due to the positive relationship between the debt ratio and the company's financial distress. As financial difficulties increase, the company faces a higher risk of bankruptcy, motivating managers to improve their performance to avoid losing their jobs or tarnishing their reputation. By minimizing agency costs between shareholders and managers, the agency cost theory advocates for the strategic implementation of debt as a means to align the interests of managers with those of shareholders and enhance overall business performance.

#### 2.2.2. Durand's Classical Theory

Durand (1952) is credited with introducing the theory of the capital structure of the firm. He argued that debt has a "cheaper" cost of capital compared to equity. Consequently, if a company utilizes a significant amount of debt, it can reduce the average cost of capital and enhance the overall value of the enterprise. Additionally, as the debt-to-equity ratio increases, the return on equity is expected to rise since the cost of equity is higher than that of debt. However, Durand also acknowledged that increasing the debt-to-total capital ratio could lead to a higher cost of debt due to the increased risk of bankruptcy. As a result, the impact of capital structure on firm value relies on achieving a balance between the advantages of using debt and equity. Therefore, companies must strive to establish an optimal capital structure that minimizes the average cost of capital and maximizes the overall value of the enterprise. It is important to note that this study refrains from drawing definitive conclusions about the precise optimal capital structure for enterprises, as it is subject to various factors and considerations specific to each business.

#### 2.2.3. The Theory of Modigliani and Miller

Modigliani and Miller (1958) put forth the argument that if a firm employs a substantial amount of debt, its shareholders are likely to invest in shares of the company with less debt as a risk-reducing strategy. The key finding of Modigliani and Miller is that a firm's overall value is independent of its debt ratio. However, in a subsequent study conducted in 1963, they introduced new evidence indicating that the cost of capital does influence capital structure and consequently impacts the firm's value. By utilizing debt, companies incur interest expenses, which are partially deductible when calculating corporate income tax. This tax advantage, referred to as tax shields, enables businesses to lower their corporate income tax costs, resulting in an enhancement of the firm's overall value. The groundbreaking theory of Modigliani and Miller laid the groundwork for the development of subsequent theories on capital structure.

#### 2.2.4. Trade-Off Theory

The capital structure trade-off theory was initiated by (Kraus and Litzenberger 1973) and later developed by (Myers and Majluf 1984). According to this theory, there exists an optimal capital structure that enhances the firm's value while considering the costs associated with financial distress. Specifically, a company can continue to borrow until the tax benefits derived from borrowing are balanced with the increased costs resulting from potential financial difficulties. At an average level of debt, the likelihood of facing financial distress is minimal, and the present value of the costs associated with financial distress

remains relatively small. Therefore, borrowing provides advantages to the firm. However, as the level of debt increases, the risk of bankruptcy also rises, which can ultimately decrease the firm's overall value. The trade-off theory of capital structure provides insights into the variations in capital structure among different types of businesses and suggests a tendency for companies to aim for an optimal capital structure based on their performance and risk tolerance.

### 2.2.5. Pecking Order Theory

The pecking order theory, proposed by (Myers and Majluf 1984), posits that firms have a preference for internal financing over external financing. According to this theory, businesses prioritize the use of retained earnings for investments, and only resort to external financing when necessary. In the pecking order, firms will first borrow, then issue bonds, and as a last resort, issue shares. The pecking order theory does not invalidate the arguments made by previous theories regarding the importance of tax shields and financial constraints on debt. Instead, it emphasizes that the order in which firms access funding sources is more significant than these factors. A well-performing business typically carries little debt, not due to the fact that it has a low optimal debt ratio, but since it relies on its internal funds and does not require external financing. On the other hand, businesses with poor performance often have high levels of debt since they lack sufficient internal funding for their projects and must seek external financing to sustain their operations. As a result, the observed debt ratio of each firm reflects the cumulative need for external financing that has built up over time in response to their performance and funding requirements.

### 2.3. Hypotheses and Research Model

#### 2.3.1. Research Hypothesis

The effect of the ratio of liabilities-to-assets (Lia) on the value of the company (ROA, ROE, and Tobin's Q):

The research results give different conclusions about the impact of the debt ratio on the firm value. However, the general trend recorded that the effect of financial leverage is to take advantage of the tax shield, making the average cost of capital of the enterprise low (Modigliani and Miller 1963), thereby increasing the value of ROA, ROE, and Tobin's Q. Combined with the characteristics of Vietnamese enterprises, the debt-to-assets structure is at 43% < 50%; therefore, it is quite safe. We expect an increase in the debt ratio to have a positive impact on firm value. Inheriting the studies of (Khan et al. 2021; Ayuba et al. 2019; Natsir and Yusbardini 2020; Ramli et al. 2019; Hirdinis 2019; Nenu et al. 2018; Aggarwal and Padhan 2017; Titman and Wessels 1988; Friend and Lang 1988; Berger and Bonaccorsi Di Patti 2006; Weill 2008; Chowdhury and Chowdhury 2010; Sivathaasan et al. 2013; Khidmat and Rehman 2014) on the positive impact of capital structure on firm value, the authors build hypotheses H1, H2, and H3 as follows:

- H1: Liabilities-to-assets ratio (Lia) has a positive effect on ROA.
- H2: Liabilities-to-assets ratio (Lia) has a positive effect on ROE.
- H3: Liabilities-to-assets ratio (Lia) has a positive effect on Tobin's Q.

The effect of long-term debt-to-assets (Llia) on firm value (ROA, ROE, and Tobin's Q):

According to (Titman and Wessels 1988), long-term loans play a vital role as capital sources for financing production and business activities. These loans are commonly obtained through bank loans or bond issuance. Bank loans usually require collateral, while bonds necessitate a good market reputation. Larger enterprises typically have the capacity to issue bonds to raise significant capital volumes, with these bonds often being guaranteed by investment banks or securities companies. In normal circumstances, businesses should leverage long-term loans to finance their operations. However, Vietnamese enterprises currently allocate only about 6% of total assets to long-term loans, indicating a relatively low utilization rate. (Abor 2005) found that the long-term debt ratio negatively correlates with certain performance indicators like ROE. Conversely, the research of (Chowdhury and Chowdhury 2010; Berger and Bonaccorsi Di Patti 2006; Sivathaasan et al. 2013; Weill 2008;

Khidmat and Rehman 2014; Titman and Wessels 1988; Friend and Lang 1988) supports the notion that long-term loans have a positive impact on firm value. Therefore, we inherit these studies and build hypotheses H4, H5, and H6 as follows:

- H4: Long-term loan-to-assets ratio (Llia) has a positive effect on ROA.
- H5: Long-term loan-to-assets ratio (Llia) has a positive effect on ROE.
- H6: Long-term loan-to-assets ratio (Llia) has a positive effect on Tobin's Q.

The effect of short-term and long-term debt-to-assets ratio (Tlia) has a negative impact on firm value (ROA, ROE, and Tobin's Q):

The trade-off theory proposed by (Kraus and Litzenberger 1973) suggests that increasing the long-term debt ratio to benefit from tax shields can also improve a firm's profitability. Enterprises listed on the Vietnamese stock market have maintained the ratio of long-term and short-term debt-to-total assets at less than 19%, which implies that increasing debt leverage through the ratio of short-term and long-term debt can enhance operational efficiency. However, numerous studies have shown that higher debt levels increase the risk of bankruptcy, leading to reduced firm performance, especially in emerging economies. Empirical research in Vietnam provides evidence of the negative impact of debt ratio on firm profitability, which depends on the economic context and current financial capacity of businesses. In the global context, studies conducted by (Nguyen et al. 2023; Masulis 1983; Singh and Faircloth 2005; Bolek and Wilinski 2012; Seetanah et al. 2014; Dawar 2014; Zeitun and Haq 2015; Minnema and Andersson 2018; Owolabi and Obida 2012; Lazaridis and Tryfonidis 2006; Liargovas and Skandalis 2008) have all confirmed the negative impact of both short-term and long-term loans on firm value. Therefore, we build hypotheses H7, H8, and H9 as follows:

- H7: Short-term and long-term borrowings-to-assets ratio (Tlia) has a negative impact on ROA.
- H8: Short-term and long-term borrowings-to-assets ratio (Tlia) has a negative impact on ROE.
- H9: Short-term and long-term borrowings-to-assets ratio (Tlia) has a negative impact on Tobin's Q.

The effect of firm size on firm value:

Large-scale enterprises often have a long process of formation and development, as well as high credibility in the market. Therefore, these businesses have more advantages in borrowing from banks. Large enterprise size leads to higher agency costs and debts have proven to be a tool to minimize conflicts of interest between shareholders and managers, motivating managers to operate for the purpose of maximizing interests (Berle and Means 1932). The studies of (Titman and Wessels 1988; Antoniou et al. 2008) all provide evidence to support the positive relationship between firm size and capital structure. Inheriting these results, the authors build hypothesis H2 as follows:

- H10: Firm size has a positive effect on firm value (ROA, ROE, and Tobin's Q).

2.3.2. Research Model

Based on the literature review and the adoption of the (Easton and Harris 1991) model to examine empirical data collected on the Vietnam Stock Exchange, three regression models are represented as follows:

$$\text{ROA} = \beta_0 + \beta_1 \text{Lia it} + \beta_2 \text{LLia it} + \beta_3 \text{TLia it} + \beta_4 \text{Size it} + \quad \varepsilon_{it} \tag{1}$$

$$\text{ROE} = \alpha_0 + \alpha_1 \text{Lia it} + \alpha_2 \text{LLia it} + \alpha_3 \text{TLia it} + \alpha_4 \text{Size it} + \quad \varepsilon_{it} \tag{2}$$

$$\text{Tobin's Q} = \lambda_0 + \lambda_1 \text{Lia it} + \lambda_2 \text{LLia it} + \lambda_3 \text{TLia it} + \lambda_4 \text{Size it} + \quad \varepsilon_{it} \tag{3}$$

where

it = Observed variable of company i at time t;
i = 1, 2, . . . 769 companies;
t = 2012, 2013, . . . 2022 (11 years total).
Defining variables in the model are detailed in Table 1 below.

**Table 1.** Defining variables in the model.

| Variable Code | Variable Name | Deterministic Formula | Expectations |
| --- | --- | --- | --- |
| Variable dependencies | | | |
| ROA | Return on assets | Profit after tax/Average assets | + |
| ROE | Return on equity | Profit after tax/Average Equity | + |
| Tobin's Q | Tobin's Q | Field value of the business/Book value of the business | + |
| Independent variable | | | |
| Lia | Debt ratio | Liabilities/Assets | + |
| LLia | Long-term loan coefficient | Long-Term Loans/Assets | - |
| TLia | Short-term and long-term loan coefficients | Total Short- and Long-Term Loans/Assets | + |
| Size | Enterprise size | Natural logarithm of total assets | + |

Source: General Author.

## 3. Research Methods

### 3.1. Research Data

The study used secondary data collected from audited financial statements of 769 companies listed on the Ho Chi Minh and Hanoi Stock Exchange in the period from 2012 to 2022 from the data platform of FiinPro Joint Stock Company. The total number of observations was 8459 (769 enterprises × 11 years).

### 3.2. Data Analysis

The authors conducted normality tests on the analyzed dependent variables (ROA, ROE, and Tobin's Q) in Stata to ensure that they followed a normal distribution before running OLS regressions.

The study used tabular data for the established econometric model to examine the impact of capital structure on the value of enterprises listed on the Vietnam Stock Exchange. The regression methods applied include the usual OLS method, the FEM regression method, and the stochastic effect model scale method. After using a regression method suitable for the model, the authors conduct a reliability test of the selected model. In the case that any model has violated the defects, the authors apply the least squared generalization (GLS) regression method to overcome the above defects.

## 4. Research Results

### 4.1. Descriptive Statistics for the Mean Values

Firm value: The statistical data in Table 2 show that the average return on assets (ROA) is 0.05 or 5.30%, with the lowest value of −0.62 and the highest value of 0.83. The average return on equity (ROE) is 0.10, where the minimum is −7.50 and the maximum is 2.93. Regarding Tobin's Q indicator, the average value is 1.19 while the highest is 31.5.

Debt structure: Debt ratio is studied through the following three criteria: Overall debt ratio (Lia), long-term debt ratio (Llia), and debt ratio (Tlia), with the average values of 0.43, 0.06, and 0.19, respectively. Thus, the listed companies maintain a fairly safe debt structure, with liabilities accounting for 43% of total assets (long-term debts only 6%), total short-term and long-term debts accounting for 19% of total assets (short-term 13%). With the debt structure (<0.5), it proves that the enterprises keep the debt level quite safe for

their financial situation. Since the listed companies are often large with strong financial potential, the ability to access traditional capital sources, such as debt or stock issuance, is favorable.

**Table 2.** Description of variables.

| Variable | Obs | Mean | Std. Dev. | Min | Max |
|---|---|---|---|---|---|
| ROA | 8459 | 0.053028 | 0.076065 | −0.62458 | 0.839056 |
| ROE | 8459 | 0.103189 | 0.18926 | −7.50341 | 2.93092 |
| Tobin's Q | 8459 | 1.188016 | 1.302597 | 0.0500 | 31.50883 |
| Lia | 8459 | 0.433137 | 0.259052 | 0.0100 | 1.294471 |
| Llia | 8459 | 0.060658 | 0.112753 | 0.0120 | 0.784767 |
| Tlia | 8459 | 0.191507 | 0.188421 | 0.0131 | 0.864737 |
| Tổng tài sản | 8459 | 10,940,805,914,099 | | 9,167,386,266 | 2,120,527,690,000,000 |

Source: Data processing results from Stata software.

Firm size: Enterprise size (size) is measured through total assets. Table 2 shows that the average total assets is 10,490 billion VND, the minimum value is 9.1 billion VND (LICOGI 14 company in 2012), and the maximum value is 2120 billion VND (VICASA Steel Company—VNSTEEL in 2022).

Firm value: Table 3 shows that firm value measured by return on assets (ROA) is relatively stable over the years from 2012 to 2022, ranging from 0.04 to 0.06, in which 2012 recorded the lowest ROA (0.04) while 2015 recorded the highest (0.06). Regarding the rate of return on equity (ROE), the average value of the period from 2012 to 2022 ranges from 0.07 to 0.12, in which 2012 and 2022 recorded the lowest ROE ratio (0.07), while the highest ROE is in 2015 and 2017 (0.12). For firm value calculated by Tobin's Q coefficient, the average value is 1.10–1.24, in which Tobin's Q recorded the largest in 2016 (1.24) and the lowest is in 2012 (1.10).

**Table 3.** Mean values of variables.

| Year | Firm Value | | | Debt Structure | | | Total Assets |
|---|---|---|---|---|---|---|---|
| | ROA | ROE | Tobin's Q | Lia | Llia | Tlia | |
| 2012 | 0.046 | 0.0743 | 1.1 | 0.41 | 0.06 | 0.18 | 6,585,641,398,696 |
| 2013 | 0.0442 | 0.0891 | 1.16 | 0.41 | 0.06 | 0.18 | 7,330,226,522,452 |
| 2014 | 0.054 | 0.1139 | 1.19 | 0.43 | 0.07 | 0.19 | 8,125,962,605,026 |
| 2015 | 0.0618 | 0.1221 | 1.18 | 0.43 | 0.07 | 0.19 | 9,309,743,234,160 |
| 2016 | 0.0585 | 0.1168 | 1.24 | 0.43 | 0.07 | 0.2 | 10,604,223,700,000 |
| 2017 | 0.0617 | 0.1248 | 1.22 | 0.44 | 0.06 | 0.2 | 12,383,069,600,000 |
| 2018 | 0.06 | 0.1169 | 1.23 | 0.45 | 0.06 | 0.2 | 13,472,626,100,000 |
| 2019 | 0.0528 | 0.1085 | 1.2 | 0.45 | 0.06 | 0.2 | 15,264,947,300,000 |
| 2020 | 0.046 | 0.0916 | 1.2 | 0.45 | 0.06 | 0.2 | 16,828,819,300,000 |
| 2021 | 0.0536 | 0.0997 | 1.19 | 0.45 | 0.06 | 0.2 | 19,488,520,300,000 |
| 2022 | 0.0447 | 0.0774 | 1.16 | 0.42 | 0.05 | 0.18 | 23,253,861,900,000 |

Source: Data processing results from Stata software.

In general, 2012 was the year when the ROA, ROE, and Tobin's Q indexes had the lowest values, since it was the period when Vietnam's economy was affected by the global economic crisis. For this reason, starting from 2012, the Government of Vietnam had to implement synchronously strong solutions to stabilize the macro-economy and

restructure the economy, giving priority to tightening monetary policy to control inflation. Therefore, from 2015 onwards, the economy in general has been restored, and thus ROA, ROE, and Tobin's Q of enterprises all tended to increase steadily. By 2020–2021, the economy of Vietnam and the world were affected by the COVID-19 epidemic; therefore, the indicators of business performance were low, especially in the service, transportation, and aviation industries.

Debt structure: Table 3 shows the fluctuations of the indicators for the period 2012–2022, showing that the overall debt ratio (Lia) has a stable level, which is 0.4 (liabilities account for 40% of total assets). The long-term debt ratio (Llia) also has a stable average value of 0.06 (equivalent to 6% of long-term debt in total assets). The debt ratio (Tlia) including short-term and long-term loans accounts for an average of 0.18–0.20.

Firm size: Considering the whole period from 2012 to 2022, the data in Table 3 show that the average value of total assets increases steadily from 6585 billion in 2012 to 23,253 billion in 2022.

In summary, although the firm value is affected by many different objective and subjective reasons, the overview of the annual data shows that, except for large fluctuations from the macro economy, ROA, ROE, and Tobin's Q all tended to increase, while the proportion of financing from liabilities decreased gradually.

Table 4 shows the following: Model 1—the correlation matrix between the independent variables and ROA confirmed that the variables Lia, Llia, and Tlia are all negatively correlated with the dependent variable ROA. Model 2—The correlation between the independent variables and ROE shows that the variables Llia and Tlia are negatively correlated with the dependent variable ROA, except for the variable Lia which has a positive correlation. Model 3—The correlation matrix table between the independent variables and Tobin's Q shows that the variables Lia, Llia, and Tlia are positively correlated with the dependent variable Tobin's Q. In all three models, the variable of firm size (LnTTS) is not correlated with ROA, ROE, and Tobin's Q due to the fact that the coefficient sig. are all greater than 0.05. The independent variables are correlated with each other, but the correlation coefficients between the independent variables are small; therefore, the possibility of multicollinearity between the independent variables is low.

**Table 4.** Correlation matrix.

| Model 1 | ROA | Lia | Llia | Tlia | LnTTS |
|---|---|---|---|---|---|
| ROA | 1 | | | | |
| Lia | −0.1542 | 1 | | | |
| | 0.0000 | | | | |
| Llia | −0.1046 | 0.3662 | 1 | | |
| | 0.0000 | 0.0000 | | | |
| Tlia | −0.1809 | 0.0909 | 0.0078 | 1 | |
| | 0.0000 | 0.0000 | 0.0000 | | |
| LnTTS | −0.0036 | 0.0194 | −0.0733 | −0.0294 | 1 |
| | 0.7529 | 0.0861 | 0.0000 | 0.0092 | |
| **Model 2** | **ROE** | **Lia** | **Llia** | **Tlia** | **LnTTS** |
| ROE | 1 | | | | |
| Lia | 0.0171 | 1 | | | |
| | 0.115 | | | | |
| Llia | −0.0325 | 0.3662 | 1 | | |
| | 0.0028 | 0.0000 | | | |

**Table 4.** *Cont.*

| Model 2 | ROE | Lia | Llia | Tlia | LnTTS |
|---|---|---|---|---|---|
| Tlia | −0.0423 | 0.0909 | 0.0078 | 1 | |
| | 0.0001 | 0.0000 | 0.0000 | | |
| LnTTS | −0.0054 | 0.0194 | −0.0733 | −0.0294 | 1 |
| | 0.6311 | 0.0861 | 0.0000 | 0.0092 | |
| **Model 3** | **Tobin's Q** | **Lia** | **Llia** | **Tlia** | **LnTTS** |
| TobinQ | 1 | | | | |
| Lia | 0.1057 | 1 | | | |
| | 0.0000 | | | | |
| Llia | 0.1021 | 0.3662 | 1 | | |
| | 0.0000 | 0.0000 | | | |
| Tlia | 0.0474 | 0.0909 | 0.0078 | 1 | |
| | 0.0000 | 0.0000 | 0.0000 | | |
| LnTTS | −0.0064 | 0.0194 | −0.0733 | −0.0294 | 1 |
| | 0.5704 | 0.0861 | 0.0000 | 0.0092 | |

Source: Data processing results from Stata software.

In Table 5, the multicollinearity test with the coefficient of variance shows that the mean VIF of the variables in the model is 2.06 < 10. According to (Baltagi 2008), no multicollinearity occurs when the VIF coefficient < 10 for the model with secondary data.

**Table 5.** Multicollinearity test.

| Variable | VIF | 1/VIF |
|---|---|---|
| Tlia | 2.65 | 0.377312 |
| Lia | 1.93 | 0.518105 |
| Llia | 1.6 | 0.625085 |
| Mean VIF | 2.06 | |

Source: Data processing results from Stata software.

### 4.2. Regression Results

First, the author employs the ordinary least squares (OLS) method for panel data regression. However, after conducting defect tests on the OLS model, it becomes evident that some defects still persist. The value of the F-test shows that Prob > F = 0.0000 < $\alpha$ = 5%; therefore, at 5% significance level, we reject H0, namely, the data collected depends on the existence of a fixed effect in each firm over time. This shows that the FEM regression model is more suitable than the OLS regression model. Therefore, the author implements the fixed effects model (FEM) and the random effects model (REM). From the results of running the FEM and REM, Hausman's test is used to compare these two regression models. The Hausman test results are shown in the following tables, showing that Prob > chi2 = 0.0000 <= 5%; therefore, hypothesis H1 is accepted. In this case, the fixed effect estimate (FEM) is more suitable than the random effect estimate (REM), and thus the FEM model is chosen.

After selecting the FEM model, the author tests the defects of the model, such as multicollinearity test gives the results that the VIF coefficients are all < 4; therefore, there is no multicollinearity phenomenon. Modified Wald are used to test the heteroscedasticity of the FEM model. We set up two hypotheses:

**H0:** *The FEM model does not have heteroscedasticity.*

**H1:** *The FEM model has heteroscedasticity.*

The results show that the p-values are all equal to 0.0000 < α (5%), rejecting the H0 hypothesis and accepting the H1 hypothesis, proving that the FEM model has heteroscedasticity.

We perform the Wooldridge test to check for autocorrelation. The results show that the FEM model has Prob > F = 0.0001 < 5%; therefore, H0 is rejected, concluding that the FEM model has autocorrelation.

We overcome the errors of the FEM model by applying the GLS method, the results are as follows:

The results of Model 1, as presented in Table 6, indicate that the debt ratio (Lia) has a positive and statistically significant impact on ROA at a significance level of 1%. This means that as the debt-to-total assets ratio increases, there is an associated increase in ROA by 0.0116% for each 1% increase in the debt ratio. However, when examining the long-term debt ratio (Llia), the estimation results show no statistically significant impact on ROA. On the other hand, the ratio of total short-term and long-term loans-to-assets (Tlia) exhibits a negative impact on ROA with a 99% confidence level. This suggests that businesses should be cautious in borrowing excessive amounts of short-term debt, as it may hinder the increase in ROA. The findings from model 1 support the acceptance of hypotheses H1 and H7, while hypothesis H4 is rejected. Based on these research findings, businesses are advised to limit interest-paying loans, particularly short-term loans. Instead, they should consider increasing other funding sources, such as leveraging trade credit payable to sellers, internal payables, and other payables, as these measures can positively influence the value of ROA.

**Table 6.** Impact of capital structure on ROA (model 1).

| | OLS | FEM | REM | GLS |
|---|---|---|---|---|
| Lia | 0.00132 | 0.0350 *** | 0.0238 *** | 0.0116 *** |
| | [0.27] | [5.33] | [4.05] | [3.79] |
| Llia | −0.00593 | −0.0456 *** | −0.0342 *** | −0.0103 |
| | [−0.49] | [−3.26] | [−2.64] | [−1.40] |
| Tlia | −0.0592 *** | −0.0697 *** | −0.0664 *** | −0.0331 *** |
| | [−6.66] | [−6.55] | [−6.80] | [−6.33] |
| _cons | 0.0680 *** | 0.0566 *** | 0.0602 *** | 0.0455 *** |
| | [42.30] | [27.50] | [23.33] | [35.82] |
| N | 8459 | 8459 | 8459 | 8459 |
| R-sq | 0.047 | 0.027 | | |
| F-test | F(3, 8455) = 100.34 | F(3, 7687) = 47.32 | | |
| | Prob > F = 0.0000 | Prob > F = 0.0000 | | |
| LM Test | | | Wald chi2(3) = 168.64 | Wald chi2(3) = 253.58 |
| | | | Prob > chi2 = 0.0000 | Prob > chi2 = 0.0000 |
| Hausman Test | | chi2(3) = 19.58 | | |
| | | Prob > chi2 = 0.0002 | | |
| Modified Wald Test | | chi2(769) = $2.0 \times 10^{10}$ | | |
| | | Prob > chi2 = 0.0000 | | |
| Wooldrige test | | F(1, 768) = 27.970 | | |
| | | Prob > F = 0.0000 | | |
| White's Test | Chi2(9) = 205.60 | | | |
| | Prob > F = 0.0000 | | | |

t statistics in brackets

*** $p < 0.01$

Source: Data processing results from Stata software.

Regression results in Table 7 show that debt structure has an impact on the dependent variable ROE with 99% confidence. As a result, the debt-to-assets ratio (Lia) and long-term debt-to-assets ratio (Llia) have a positive impact on ROE, and the impact level of the overall debt ratio (Lia) is higher. This result encourages businesses to increase loans, especially long-term debt. For short-term and long-term debt-to-asset ratios, there is a negative impact on ROE with an impact level of −0.0917. Research results prove that enterprises should not use short-term debt since it causes ROE to decrease. Thus, the increase in short-term debt will reduce the operating efficiency of the enterprise in terms of ROE. Then, hypotheses H2, H5, and H8 are accepted.

**Table 7.** Impact of capital structure on ROE (model 2).

|  | OLS | FEM | REM | GLS |
|---|---|---|---|---|
| Lia | 0.187 *** | 0.258 *** | 0.217 *** | 0.167 *** |
|  | [15.67] | [13.19] | [14.56] | [21.89] |
| Llia | 0.116 *** | 0.142 *** | 0.137 *** | 0.0867 *** |
|  | [3.92] | [3.40] | [3.98] | [3.75] |
| Tlia | −0.202 *** | −0.254 *** | −0.224 *** | −0.0917 *** |
|  | [−9.36] | [−8.03] | [−8.73] | [−6.14] |
| _cons | 0.0764 *** | 0.0580 *** | 0.0687 *** | 0.0566 *** |
|  | [19.54] | [9.47] | [12.55] | [28.03] |
| N | 8459 | 8459 | 8459 | 8459 |
| R-sq | 0.09 | 0.116 |  |  |
| F-test | F(3, 8455) = 16.77 | F(3, 7687) = 13.38 |  |  |
|  | Prob > F = 0.0000 | Prob > F = 0.0000 |  |  |
| LM Test |  |  | Wald chi2(3) = 43.29 | Wald chi2(3) = 219.87 |
|  |  |  | Prob > chi2 = 0.0000 | Prob > chi2 = 0.0000 |
| Hausman Test |  | chi2(3) = 6.38 |  |  |
|  |  | Prob > chi2 = 0.0000 |  |  |
| Modified Wald Test |  | chi2(769) = $5.5 \times 10^{0.7}$ |  |  |
|  |  | Prob > chi2 = 0.0000 |  |  |
| Wooldrige test |  | F(1, 768) = 2.298 |  |  |
|  |  | Prob > F = 0.0000 |  |  |
| White's Test | Chi2(9) = 170.40 |  |  |  |
|  | Prob > F = 0.0001 |  |  |  |

t statistics in brackets

*** $p < 0.01$

Source: Data processing results from Stata software.

Table 8 shows that the estimated results of model 3 (Tobin's Q) have similar results with model 1 (ROA). Debt ratio (Lia) has a positive effect on Tobin's Q at 1% significance level. This proves that the higher the debt ratio, the more Tobin's Q increases. Specifically, when the debt-to-total assets ratio increases by 1%, Tobin's Q increases by 0.450%, which is a fairly large influence. Taking a closer look at long-term loans through the long-term debt coefficient (Llia), the model 3 estimation results show that there is no statistically significant impact of long-term loans on Tobin's Q. For the coefficient of total short-term and long-term loans-to-assets (Tlia), there is a negative impact on Tobin's Q with 99% confidence. This proves that businesses should not borrow a large amount of short-term debt, since it has a restraining effect on the growth of Tobin's Q. The estimation results of model 3 also affirm

that hypotheses H3 and H9 are accepted, and hypothesis H6 is not accepted. From the research results, businesses should limit interest-paying loans, especially short-term loans. They should pay attention to increasing other amounts, such as taking advantage of trade credit, payable to sellers, taxes and other payables, payable to employees, payable expenses, internal payables, etc., as these will have an effect on increasing the value of Tobin's Q.

**Table 8.** Impact of capital structure on Tobin's Q (model 3).

| | OLS | FEM | REM | GLS |
|---|---|---|---|---|
| Lia | 0.526 *** | 1.259 *** | 1.164 *** | 0.450 *** |
| | [6.30] | [16.41] | [15.69] | [15.61] |
| Llia | 1.846 *** | 0.0517 | 0.229 | 0.025 |
| | [8.88] | [0.32] | [1.43] | [0.38] |
| Tlia | −1.592 *** | −0.770 *** | −0.842 *** | −0.562 *** |
| | [−10.53] | [−6.19] | [−6.94] | [−12.10] |
| _cons | 1.027 *** | 0.634 *** | 0.679 *** | 0.678 *** |
| | [37.49] | [26.35] | [14.97] | [64.58] |
| N | 8459 | 8459 | 8459 | 8459 |
| R-sq | 0.058 | 0.224 | | |
| F-test | F(3, 8455) = 65.98 | F(3, 7687) = 241.65 | | |
| | Prob > F = 0.0000 | Prob > F = 0.0000 | | |
| LM Test | | | Wald chi2(3) = 672.85 | Wald chi2(3) = 1726.33 |
| | | | Prob > chi2 = 0.0000 | Prob > chi2 = 0.0000 |
| Hausman Test | | chi2(3) = 68.98 | | |
| | | Prob > chi2 = 0.0000 | | |
| Modified Wald Test | | chi2(769) = 1.75 × $10^{0.7}$ | | |
| | | Prob > chi2 = 0.0000 | | |
| Wooldrige test | | F(1, 768) = 7.401 | | |
| | | Prob > F = 0.0067 | | |
| White's Test | Chi2(9) = 34.95 | | | |
| | Prob > F = 0.0001 | | | |
| | t statistics in brackets | | | |
| | *** $p < 0.01$ | | | |

Source: Data processing results from Stata software.

Specifically, the results of testing the hypotheses are as follows:

Hypotheses H1, H2, and H3: Debt-to-assets ratio (Lia) has a positive impact on firm value as measured through ROA, ROE, and Tobin's Q with 99% confidence. In particular, the impact of the overall debt coefficient on Tobin's Q is the largest (0.450), followed by ROE (0.167), and finally ROA (0.0116). This proves that the market is willing to pay high prices for companies that use large financial leverage in their business activities. Therefore, it does not contradict the results about the impact of debt-to-total assets on ROA and ROE indicators when owners and shareholders may still prefer to increase the debt ratio to finance business operations. This result supports hypotheses H1, H2, and H3 as well as the prioritization theory. The issuance of more shares to increase equity and reduce the debt structure will reduce the company's stock market value. In particular, the issuance of more stocks by the company will lead to the dilution of stocks and reduction in the market value of the company, since the market will not appreciate these companies. This result

agrees with the studies of (Khan et al. 2021; Ayuba et al. 2019; Natsir and Yusbardini 2020; Ramli et al. 2019; Hirdinis 2019; Nenu et al. 2018; Aggarwal and Padhan 2017; Chowdhury and Chowdhury 2010; Khidmat and Rehman 2014; Titman and Wessels 1988; Friend and Lang 1988; Sivathaasan et al. 2013; Tran 2016; Đ. B. Thanh 2016; Vo 2017 and contrasted with Masulis 1983; Singh and Faircloth 2005; Bolek and Wilinski 2012; Seetanah et al. 2014).

Hypotheses H4, H5, and H6: Regression results are quite diverse on the impact of long-term debt-to-assets ratio (Llia) on firm value. First, long-term debt-to-total assets ratio (Llia) has a positive effect on ROE at 1% significance level and supports hypothesis H5. The ratio of long-term debt-to-total assets does not affect ROA and Tobin's Q; therefore, hypotheses H4 and H6 are not accepted. Although, the impact of long-term debt structure (0.0867) is much smaller than that of the debt ratio in general (0.167). This indicates a strong demand among listed companies for long-term loans, and the potential for increased profitability is closely linked to their ability to secure long-term capital. In essence, the composition of long-term loans plays a critical role in enhancing the profitability of these listed companies, particularly their return on equity (ROE). The market seems to favor businesses that utilize significant leverage to finance their operations, a finding consistent with the studies of (Khidmat and Rehman 2014; Titman and Wessels 1988; Friend and Lang 1988; Sivathaasan et al. 2013; Dawar 2014; Zeitun and Haq 2015; Minnema and Andersson 2018; Owolabi and Obida 2012; Lazaridis and Tryfonidis 2006; Liargovas and Skandalis 2008 and in contrast to the results of Dawar 2014; Zeitun and Haq 2015; Minnema and Andersson 2018; Owolabi and Obida 2012; Lazaridis and Tryfonidis 2006; Liargovas and Skandalis 2008). Next, the long-term debt-to-assets ratio (Llia) has no impact on ROA and Tobin's Q. This result agrees with (Jiraporn and Liu 2008; Ebaid 2009; Phillips and Sipahioglu 2004; contrary to authors Titman and Wessels 1988; Friend and Lang 1988; Sivathaasan et al. 2013).

Hypotheses H7, H8, and H9: The ratio of short-term and long-term debt-to-assets (Tlia) has a negative effect on the measures of firm value (ROA, ROE, and Tobin's Q). The results are all statistically significant at 1%. Therefore, hypotheses H7, H8, and H9 are accepted. While hypothesis H4 has proven that the long-term debt ratio (Llia) does not affect the value of the enterprise, it follows that if the enterprise increases the short-term debt, it will reduce their value according to the measurement (ROA, ROE, and Tobin's Q). Specifically, a 1% increase in short-term borrowing ratio will cause Tobin's Q to decrease by 0.562%, ROA to decrease by 0.0331%, and ROE to decrease by 0.0917%. This may be due to the fact that the short-term debt structure accounts for a large proportion of the total debt structure of enterprises, in order that an increase in the proportion of short-term debt will expose them to payment risks, financial risks, as well as high capital costs. The above descriptive analysis also shows that the ratio of short-term debt-to-total assets of listed companies still tends to increase, short-term debt is three times higher than long-term debt (Table 3). Therefore, the pressure and financial risks of listed companies are quite large. This research result is consistent with previous studies of (Nguyen et al. 2023; Masulis 1983; Bolek and Wilinski 2012; Seetanah et al. 2014; Dawar 2014; Zeitun and Haq 2015; Minnema and Andersson 2018); however, it is contrary to the conclusion of (Abor 2005).

From the model estimation results in Tables 5–7, it confirms that capital structure has the most impact on Tobin's Q and the least impact on ROA. Specifically, a 1% increase in debt-to-assets ratio (Lia) caused Tobin's Q to increase strongly (0.45%), ROE to increase by 0.167%, and ROA by 0.0116%. The long-term debt ratio has no impact on firm value since the Llia coefficients are not statistically significant. In contrast, the ratio of short-term and long-term debt (Tlia) has the opposite effect, namely, when increasing by 1%, Tlia causes Tobin's Q to decrease by 0.562%, and both ROA and ROE to decrease. This suggests that enterprises should not increase the proportion of short-term loans since it does not increase the value of the enterprise. Moreover, the usage of short-term loans to finance long-term assets will lead to financial imbalance. If businesses abuse short-term loans to take advantage of tax shields, it will lead to an imbalance in the debt structure. Moreover, when using short-term loans, it will also cause pressure to pay short-term debts, increasing

the pressure of capital turnover of enterprises, which may cause difficulties for enterprises in terms of management and control stages in the production and business process, leading both efficiency and profitability to potentially decrease.

## 5. Conclusions and Recommendations

The empirical study uses a sample of 8459 observations, collected from 769 companies listed on the Vietnamese stock market from 2012 to 2022. Various regression methods are applied, including OLS, FEM, REM, and GLS. Dependent variables to measure firm value include ROA, ROE, and Tobin's Q. The independent variable is capital structure with scales including debt-to-assets (Lia), long-term debt-to-assets (llia), short-term and long-term debt-to-assets (Tlia), and firm size. The results of using the GLS model show that the debt ratio (Lia) has a positive impact on all three firm value indicators, in which the strongest impact is on Tobin's Q. The long-term debt ratio has no impact on firm value. Short-term and long-term debt ratios have negative effects on ROA, ROE, and Tobin's Q, in which the impact on Tobin's Q reduction is the most (0.562). Research results encourage businesses to use less short-term debt rather than taking advantage of other capital sources, such as commercial credit, internal loans, etc.

In general, the higher the Tobin's Q in market coefficient, the lower the benefit from debt mobilization. The increase in debt to finance business activities only has a positive effect on the market price of the business when the ratio of market value/book value is low. Thus, for enterprises with high market value, they should give priority to raising capital from stock issuance since the advantage of high stock prices can raise a large amount of capital with lower mobilization costs.

As for ROE, capital structure is one of the factors that has a significant influence on the return on equity of the enterprise. Accordingly, the profitability ratio of listed companies can be maximized when they maintain the debt-to-total assets ratio around the optimal level of 41–45%. Compared with the average debt ratio of the whole market of 63%, it shows that most enterprises are maintaining the debt ratio beyond the optimal threshold. This partly explains the negative result in the linear regression model between debt structure and return on equity. From the results, we propose the following solutions:

Recommendations related to choosing a reasonable capital structure

Research results have shown that long-term debt does not affect firm value, but short-term debt has a negative impact. This is the reason why it is recommended that businesses should not maintain a high debt ratio. Moreover, maintaining a high debt ratio makes the risk in business activities of enterprises high. Businesses may face financial distress when they cannot afford to pay their debts or can do it but find it very difficult. This situation can cause some trouble for them, but it can also lead to the possibility of the business going bankrupt. On the contrary, the issue of stocks can help public companies attract a large amount of capital to expand the scope of business activities. However, the cost to issue stocks is often high and the pressure to maintain the growth rate increases for businesses. Enterprises also face other risks such as loss of control, in which the company value decreases if it does not meet investors' expectations.

According to the agency cost theory, debt is similar to a mechanism that monitors and encourages the performance of the board of directors due to the positive relationship between debt ratio and financial difficulties of the company. However, considering the conflict between shareholders and creditors, debt has the effect of increasing agency costs. As debt levels rise, creditors tend to demand a higher interest rate on loans to compensate for the risks they may face. As each capital raising tool is issued, businesses must spend certain costs. For debt instruments, the business needs to pay interest; for equity instruments, the business needs to meet investors' expectations through the level of dividends paid or the growth of the business in the future. Therefore, the policy on capital structure needs to harmonize the interests of shareholders, the board of directors, the executive board, and creditors.

When selecting managers, it is crucial to design a capital structure tailored to the specific characteristics and developmental stage of the company. Several internal factors come into play during this planning process, including the business plan and the financial requirements to execute it, the current solvency status of the company, the interest expenses to be borne, and the anticipated profitability or dividends for shareholders. Additionally, external factors warrant careful examination, such as legal regulations governing capital issuance, the conditions and procedures for issuing new shares or bonds, and the associated issuance costs. Furthermore, enterprises should be mindful of the potential risks of being subject to takeovers or losing control to competitors. A comprehensive consideration of these factors will facilitate the creation of an optimal and well-suited capital structure for the company's sustainable growth.

Recommendations to improve the profitability of the business

In the current business landscape, the trend of diversifying into multiple industries presents promising opportunities. However, enterprises should remain focused on their core business activities to leverage their inherent strengths effectively. It is essential for firms to prioritize capital allocation to support their primary operations before considering other investment ventures. When engaging in non-core investment activities, enterprises must implement robust control measures to minimize risks and maximize potential benefits. The nature of capital invested in non-core industries should be clearly defined, ensuring that it comes from excess funds generated through production and core business activities, and its duration should be carefully assessed, distinguishing between temporary and long-term allocations. In the case where investments in non-core ventures proves ineffective, prompt measures for divestment should be taken. Regularly reviewing and evaluating the current investments of the business will allow managers to make informed decisions and optimize the overall performance of the enterprise.

Limitations of the study: This study only focused on 769 listed companies and considered business performance through three criteria: ROA, ROE, and Tobin's Q. Future studies may expand the sample range, add other financial indicators, such as ROS, ROCE, the long-term debt ratio, non-financial indicators, or study the non-linear impact of capital structure on firm value, compared by different fields, sizes, and business lines.

**Author Contributions:** Conceptualization, K.T.P.; methodology, X.H.N.; software, T.N.B. and K.T.P.; validation, T.N.B. and X.H.N.; formal analysis, T.N.B.; resources, K.T.P.; data curation, T.N.B. and K.T.P.; writing—original draft preparation, X.H.N. and K.T.P.; writing—review and editing, X.H.N.; visualization, T.N.B.; supervision, T.N.B. All authors have read and agreed to the published version of the manuscript.

**Funding:** This research received no external funding.

**Informed Consent Statement:** Not applicable.

**Data Availability Statement:** Not applicable.

**Conflicts of Interest:** The authors declare no conflict of interest.

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
