# Peer review of "The Effect of Capital Structure on Firm Value: A Study of Companies Listed on the Vietnamese Stock Market"

_ijfs, doi:10.3390/ijfs11030100_

Round 1

Reviewer 1 Report

The submitted paper presents a very interesting analysis of the impact of capital structure on a series of financial indicators, like ROA, ROE and Tobin’s Q. The topic is of high interest for research and practice. The sample data and the comparison between the paper’s results and the findings of other research represent strengths of this article.

However, there are some issues that need to be addressed and improved, in my opinion.

1.      Please read carefully all your phrases that end with “…”, because they are unfinished, and elaborate on them. Your article contains passages which were left in a draft form (for example lines 28-30, 66-67, 140-141, 144, 729-733).

2.      In line 78 you refer to “the author”. Please revise.

3.      Your data set seems to be very interesting. Please explain whether you have performed normality tests for the analyzed dependent variables (ROA, ROE and Tobin’s Q), and elaborate a bit more on this preliminary phase of your research.

4.      Please check and correct the number and title of the Table in line 447.

5.      Please also expand your references with recent studies. Most of your references are older than 5 years.

        I recommend to carefully check or consider editing of English language. There are some passages that need to be checked and rephrased. In addition to observation 1, I will mention a few passages that need rephrasing, but please keep in mind that my examples are not exhaustive: lines 75-77, 80-81, 144-147, 198-199, 370-376, 508, 747-747 and others.

Author Response

  1. Please check that all references are relevant to the contents of the manuscript. Please read carefully all your phrases that end with “…”, because they are unfinished, and elaborate on them. Your article contains passages which were left in a draft form (for example lines 28-30, 66-67, 140-141, 144, 729-733).

The Authors revised all the comments and highlighted in the manuscript

  1. In line 78 you refer to “the author”. Please revise.

The Authors revised

  1. Your data set seems to be very interesting. Please explain whether you have performed normality tests for the analyzed dependent variables (ROA, ROE and Tobin’s Q), and elaborate a bit more on this preliminary phase of your research.

The Authors performed normality tests for the analyzed dependent variables (ROA, ROE and Tobin’s Q) in Stata and sure that is assumed to be normally distributed when we run OLS regressions. We explained more in the method section.

  1. Please check and correct the number and title of the Table in line 447.

The Authors revised

  1. Please also expand your references with recent studies. Most of your references are older than 5 years.

The author has expanded and updated the references in the last 5 years

  1. Comments on the Quality of English Language: I recommend to carefully check or consider editing of English language. There are some passages that need to be checked and rephrased. In addition to observation 1, I will mention a few passages that need rephrasing, but please keep in mind that my examples are not exhaustive: lines 75-77, 80-81, 144-147, 198-199, 370-376, 508, 747-747 and others.

The Authors revised

Reviewer 2 Report

The paper examines the relationship between capital structure and firm value for companies listed on the Vietnamese stock market. While the study utilizes a sizable dataset spanning a ten-year period, there are several areas where the paper could be improved to enhance its contribution to the field.

The choice of estimation methods, including OLS, FEM, REM, and GLS, demonstrates a comprehensive approach. However, the paper lacks a thorough discussion on the appropriateness of these methods for capturing the complexities of the Vietnamese stock market. While the research findings suggest a positive impact of the debt ratio on financial performance indicators, such as ROA, ROE, and Tobin's Q, the magnitudes of these effects are questionable, regarding the practical significance of the observed relationships. Furthermore, the lack of a significant impact of the long-term debt ratio on firm value contradicts established theories and empirical evidence, warranting further investigation and discussion.

need to be improved

Author Response

Response to Reviewer 2 Comments

  1. The paper examines the relationship between capital structure and firm value for companies listed on the Vietnamese stock market. While the study utilizes a sizable dataset spanning a ten-year period, there are several areas where the paper could be improved to enhance its contribution to the field.

We talked about this in the section on limitations and future research directions.

We will take the comment of the reviewer and implement it in future studies

  1. The choice of estimation methods, including OLS, FEM, REM, and GLS, demonstrates a comprehensive approach. However, the paper lacks a thorough discussion on the appropriateness of these methods for capturing the complexities of the Vietnamese stock market. While the research findings suggest a positive impact of the debt ratio on financial performance indicators, such as ROA, ROE, and Tobin's Q, the magnitudes of these effects are questionable, regarding the practical significance of the observed relationships. Furthermore, the lack of a significant impact of the long-term debt ratio on firm value contradicts established theories and empirical evidence, warranting further investigation and discussion.

We have corrected it in the discussion of the research results

Reviewer 3 Report

The capital structure of companies has been a topic of great research interest, both academically and socially, and it is important and relevant in the current context. In my analysis, the paper is well-structured, with a well-written abstract, introduction, literature review, and discussion of results. The authors fulfill the purpose of an introduction by providing a general framework for the topic, justifying the choice of the theme to be analyzed, and defining the objectives to be achieved. However, they could be clearer in this aspect and in the structure of the paper, although they do present references and some concepts. They then proceed with the literature review, which I consider a crucial point in papers with similar objectives and purposes (articles).

The authors present the conclusions and evidence found by various authors. However, I believe that this is where the biggest gap in the paper lies. The authors do not reference current works adequately, as they only provide one reference from the last five years. They should add/modify the review to include recent works (from the last few years), aiming to present a ratio of at least 30% of recent works.

The methodology is presented clearly and objectively, and it is appropriate for achieving the authors' objectives. The discussion of the results is well done. The conclusion summarizes the main points of the paper and is written in an objective, simple, and highly understandable manner. It focuses on the main observations from the results obtained through the methodology used. The conclusion is balanced and concise, mentioning limitations and providing directions for future research.

Regarding the bibliography, which is a crucial aspect of a paper, I believe that the authors did not comply with the requirements. The referenced works should include at least 30% from the last five years, demonstrating the relevance and significance of the topic, and should refer to publications in highly reputable academic journals.

Author Response

Response to Reviewer 3 Comments

  1. The authors present the conclusions and evidence found by various authors. However, I believe that this is where the biggest gap in the paper lies. The authors do not reference current works adequately, as they only provide one reference from the last five years. They should add/modify the review to include recent works (from the last few years), aiming to present a ratio of at least 30% of recent works.

 The author has expanded and updated the references in the last 5 years

Round 2

Reviewer 1 Report

Dear authors, thank you for your considerable work in improving your paper.

Reviewer 3 Report

The revisions made to your article have contributed significantly to an improvement over the original work. The changes that were made have made the article much clearer and more convincing, particularly in the explanation of the tests as well as the additional references which have significantly enriched the content and strengthened the theoretical foundation.